# B³CT: THREE-BRANCH COORDINATED TRAINING FOR DOMAIN ADAPTIVE SEMANTIC SEGMENTATION

## ABSTRACT

Unsupervised domain adaptive semantic segmentation aims to adapt a dense prediction model trained on the source domain to the target domain by transferring knowledge without further annotations. A mainstream solution for transferring knowledge is to achieve alignment between different domains and eliminate domain gaps caused by source-target distributions. However, previous work paid little attention to where and when to align. We find that different contents in images are aligned at different stages of the whole network, and the alignment should be gradually strengthened during the whole training process due to the accuracy of target pseudo labels. Given these two observations, we propose a three-branch coordinated training (B³CT) framework. Besides two normal source and target branches, a third branch is involved specifically for the alignment. In this branch, the hybrid-attention mechanism is utilized to do the alignment, while an Adaptive Alignment Controller (AAC) is built to adjust the contents being aligned according to the stages of the whole network. Meanwhile, in B³CT, a coordinate weight is designed to gradually strengthen the importance of the alignment based on the training accuracy in the whole training process. Extensive experiments show that our proposed methods achieve competitive performances on tasks of GTA5→Cityscapes and SYNTHIA→Cityscapes.

## 1 INTRODUCTION

Deep neural networks have achieved remarkable success in various application scenarios, but they still suffer from expensive human labour annotation and poor adaptation performance. Thus, as a promising technique, unsupervised domain adaptation attracts much attention from academia and industry, especially for dense prediction tasks. Unsupervised domain adaptation for semantic segmentation is proposed to make semantic predictions for each pixel on the unlabeled target domain by learning a model with labeled source domain images. However, due to the significant distribution discrepancy between different domains, *i.e.*, the domain shift problem, the model trained on the source domain shows a remarkable performance drop on the target domain.

To address the domain shift problem, numerous methods are proposed to achieve feature alignment by learning domain-invariant features. Pixel-level alignment methods (Yang et al., 2020; Kim & Byun, 2020; Cheng et al., 2021; Shen et al., 2023) utilize an image translation model, such as GAN (Zhu et al., 2017), and a segmentation method iteratively to project the image styles of different domains into the same domain. Prototype-level alignment methods (Zhang et al., 2021; Liu et al., 2021a; Wang et al., 2023; Das et al., 2023) minimize distances between the class prototypes of the source and target domains. Label-level alignment methods (Tsai et al., 2018; Vu et al., 2019; Gong et al., 2023) exploit the similarity of probability and entropy to produce similar predictive distributions in the output space of the source and target domains.

However, there are still two issues exist in previous works, which are "where to align" and "when to align". For the first issue, some works (Yang et al., 2021; Wang et al., 2023; Das et al., 2023) only conduct their alignments on the high-level layers through an align loss, while some other methods (Rao et al., 2022; Wang et al., 2022; Xu et al., 2021) treat the alignment equally important in all the stages of the network, both low-level layer and high-level layer. We argue that different contents should be aligned at different stages of the network. For example, low-level information (*i.e.*, texture, edge, and color) and high-level information (*i.e.*, semantic information) should be aligned at low-

level layers and high-level layers respectively. In other words, the features being aligned should be able to be adjusted in different layers.

In this paper, we propose a three-branch coordinated training ($B^3CT$) framework, the third branch of which facilitates learning domain-invariant features for alignment. Specifically, the hybrid-attention mechanism is involved in this branch to construct feature interaction between different domains and fuse features to facilitate alignment. Based on the hybrid-attention mechanism, an Adaptive Alignment Controller (AAC) is further proposed and added into every stage of the whole network. It is used to control the contents that should be aligned at different layers. After training, with the help of AAC, the network can autonomously select which token needs to be aligned in each stage, thereby ensuring more effective knowledge transfer.

In addition, when to align features also needs to be considered. Over-training on the source domain prevents the model from learning the domain adaptive features of the target domain. On the contrary, over-training on the unlabeled target domain results in inefficient learning of discriminative features. Also, training too early on the target domain introduces noisy labels inevitably, while training too late traps the model into a local optimum, biased to the source domain feature distribution. Therefore, we take the pseudo-accuracy on the target domain as the metric and propose a coordinate weight to control the involvement of the hybrid branch in the training process.

We summarize our contributions as follows. 1) To focus on feature alignment between different domains, we propose a three-branch coordinated training ($B^3CT$) framework, which enables solving both "where to align" and "when to align" issues in a unified manner. In the third branch of the framework, the hybrid-attention is used to perform the alignment. 2) For the "where to align" issue, an Adaptive Alignment Controller (AAC) is designed to adjust the contents being aligned for each network stage during training. 3) For the "when to align" issue, we propose a coordinate weight to achieve balance training among three branches in the $B^3CT$ framework. 4) We achieve competitive performances of 74.8 on GTAV→Cityscapes task and 67.0 on SYNTHIA→Cityscapes task.

## 2 RELATED WORK

### 2.1 SEMANTIC SEGMENTATION

Semantic Segmentation aims to segment the objects and scenes in images and give their classifications. In the deep learning era, many works are based on FCN structure (Long et al., 2015), which is the fundamental work of semantic segmentation using deep learning techniques. Caused by limited receptive field and information differences in different network stages, researchers improved FCN from multiple aspects, including refining the contextual information (Yuan et al., 2018; 2020; Poudel et al., 2018), enlarging the receptive field (Zhao et al., 2017; Yang et al., 2018), or introducing attention mechanism (Fu et al., 2019; Wang et al., 2018; Zhao et al., 2018) in various ways. In recent years, many works have proposed replacing Convolutional Neural Networks (CNNs) with Vision Transformers (ViTs) (Dosovitskiy et al., 2020) in semantic segmentation task, which specializes in long-range dependency modeling, is widely viewed as a promising route for further development. However, although supervised semantic segmentation has achieved impressive results in major benchmarks, there is still significant room for improvement in the model's generalization ability for UDA semantic segmentation tasks.

### 2.2 DOMAIN ADAPTIVE SEMANTIC SEGMENTATION

The main challenge of unsupervised domain adaptive semantic segmentation is the domain shift problem, due to the distribution discrepancy between the source and target domains. Thus, previous works have shown remarkable progress by achieving feature alignment and can be summarized into the following categories. Pixel-level alignment methods (Yang et al., 2020; Kim & Byun, 2020; Cheng et al., 2021; Shen et al., 2023) first transferred the image style of different domains into the same domain by a style translation model CycleGAN (Zhu et al., 2017). Then, a segmentation model is trained on domains with the translated style. Prototype-level alignment methods (Zhang et al., 2021; Liu et al., 2021a; Wang et al., 2023; Das et al., 2023) utilize the class prototype from the source and target domains to achieve feature alignment. Label-level alignment methods (Tsai et al., 2018; Vu et al., 2019; Gong et al., 2023) exploited the probability similarity and entropy similarity to generate similar prediction distributions in the output space for either source or target domains, while another perspective is to align the prediction results of different networks with multiple feature spaces (Kumar et al., 2018). Self-training methods (Araslanov & Roth, 2021; Hoyer et al., 2022a;b)

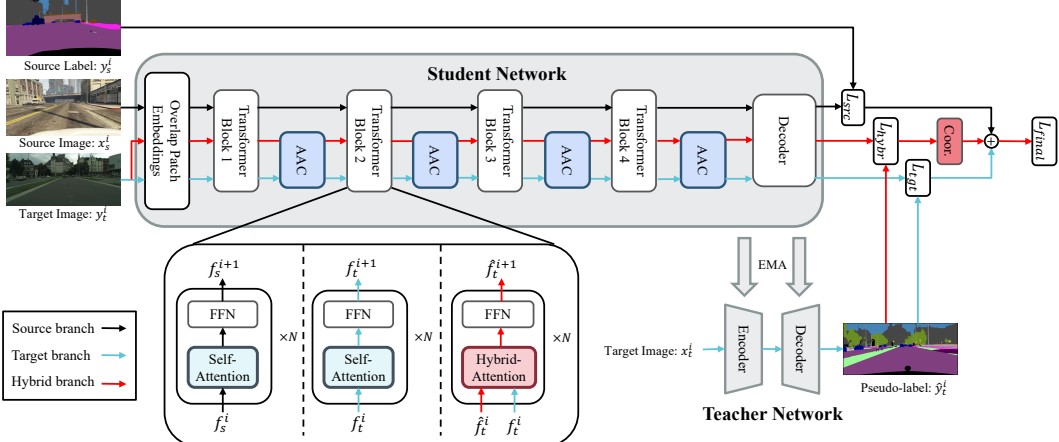

Figure 1: **Illustration of the the B³CT framework.** The B³CT framework is mainly divided into two parts: the student network and the teacher network. The teacher network is used to generate pseudo labels required for the target image of the student network training and updates the weights using EMA based on the student network weights. The student network consists of three parallel training branches, each using a different attention mechanism. The final loss function is dynamically weighted by coordinate weight on the loss values of three branches.

first generate pseudo-labels based on a pre-trained model from the source domain. Then, the model is trained on the target domain with the supervision of pseudo-labels.

## 2.3 Self attention and Cross attention

The self-attention mechanism is the core component of the Transformer (Vaswani et al., 2017). Many works (Han et al., 2020; Dosovitskiy et al., 2020; Liu et al., 2021b) have shown its effectiveness for computer-vision tasks. ViT (Dosovitskiy et al., 2020) split an image into feature tokens and took self-attention mechanism to construct a relation between feature tokens. Swin Transformer (Liu et al., 2021b) introduced the hierarchical structure into ViT (Dosovitskiy et al., 2020) and proposed a shifted windowing scheme, where self-attention is adopted within local windows for efficient computation. The cross-attention mechanism has shown great potential in feature fusion and feature alignment. Gao (Gao et al., 2019) proposed a dynamic fusion with intra-modality and inter-modality attention flow, which exploited the association weights between the visual modal and text modal on the visual question-answer task. Chen (Chen et al., 2021) designed a dual transformer architecture, where the cross-attention mechanism is adopted to exchange information between small-patch and large-patch tokens. Xu (Xu et al., 2021) introduced cross-attention into domain adaptive classification to achieve label denoising. In this paper, we take advantage of cross-attention on feature alignment to build a hybrid branch for better performance on domain adaptive semantic segmentation.

## 3 The Proposed Method

In this section, we first propose our B³CT as shown in Fig. 1, in which three weight-shared parallel branches are trained end-to-end simultaneously. In the third branch, a modified attention mechanism named hybrid-attention is utilized to perform feature alignment, which integrates intra-domain self-attention and inter-domain cross-attention mechanisms. Furthermore, we conduct an in-depth exploration of the alignment mechanism and propose an Adaptive Alignment Controller. Based on different semantic levels in different attention blocks, AAC is used to determine different degrees of alignment at each layer of the network. At last, a coordinate weight is designed to determine the involvement of the hybrid branch in the training process.

### 3.1 The three-branch coordinated training framework

Given a labeled source domain dataset $\mathbb{D}_s = \{(x_s^i, y_s^i)|y_s^i \in \mathbb{R}^{H \times W}\}_{i=1}^{N_s}$ and an unlabeled target domain dataset $\mathbb{D}_t = \{x_t^i\}_{i=1}^{N_t}$, unsupervised domain adaptive semantic segmentation predicts pixel-level semantic masks for target domain images, where $N_s$ and $N_t$ are the numbers of training data

of source and target domains respectively. The height and width of the input are denoted as $H$ and $W$. Due to the superior performance and training complexity, we follow the self-training framework (Zou et al., 2018; Wang et al., 2021b; Zhang et al., 2021; Zheng & Yang, 2021; Araslanov & Roth, 2021) and adopt the mean teacher model (Araslanov & Roth, 2021; Hoyer et al., 2022a) to achieve an end-to-end self-training learning process, avoiding the cumbersome iterative training stage (Zhang et al., 2021; Araslanov & Roth, 2021).

Specifically, our framework consists of three weight-shared parallel branches, named source branch, target branch, and hybrid branch. In the source branch, the source images are fed into a self-attention transformer to obtain the predicted category of every pixel. The cross-entropy loss is used to guide training on the source domain:

$$L_s^i = -\frac{1}{HW} \sum_{j=1}^{HW} \sum_{c=1}^{C} y_s^{i,j,c} \log g(x_s^{(i)})^{(j,c)} \tag{1}$$

where $C$ is the number of semantic categories. $g$ represents the forward function of the student network using self-attention only.

We follow the self-training training pipeline. The pseudo-labels $\hat{y}_t$ are generated online by teacher network. After the pseudo-labels are selected using a confidence threshold, the cross-entropy loss of the target domain is as follows:

$$L_t^i = -\frac{1}{HW} \sum_{j=1}^{HW} \sum_{c=1}^{C} \hat{y}_t^{i,j,c} \log g(x_t^{(i)})^{(j,c)} \tag{2}$$

The mean teacher model jointly trains the images of the source and target domains end-to-end. On the basis of the existing mean teacher model (Hoyer et al., 2022a;b), we add an additional branch that uses both source domain images and target domain images for hybrid computation in a forward propagation process, which we called the hybrid branch. Three branches shared the same weights, but used different attention mechanisms, as shown in Fig. 1.

Taking a pair of augmented source and target images, the student network first downsamples and reshapes the inputs (features) into a sequence of source tokens $f_s \in \mathbb{R}^{N \times d}$ and target tokens $f_t \in \mathbb{R}^{N \times d}$. The number of tokens in the $i$-th stage of the Transformer architecture (Xie et al., 2021) is denoted by $N = \frac{H}{2^{i+1}} \times \frac{W}{2^{i+1}}$, and $d$ is the number of channels. Then, these tokens are sent into hybrid-attention layers to propagate and aggregate information.

To explicitly construct the interaction between the source and target features during the feature learning process, borrow the idea from some cross-modal works (Saharia et al., 2022; Chen et al., 2021), we use a hybrid-attention mechanism to achieve feature alignment by fusing value features from both source and target domains adaptively. The hybrid-attention first concatenates key features from two domains as hybrid-domain key $K_h = [K_s; K_t] \in \mathbb{R}^{2N \times d_k}$ and constructs hybrid-domain value $V_h = [V_s; V_t] \in \mathbb{R}^{2N \times d_v}$. Then, considering that the goal of UDA tasks is target domain performance, source query $Q_s$ is multiplied with hybrid-domain key $K_h$ to generate a similarity matrix, which guides the linear weighted summation of hybrid-domain value $V_h$:

$$Attn_{hybrid}(Q_t, K_s, K_t, V_s, V_t) = Softmax(\frac{Q_t[K_s; K_t]^\top}{\sqrt{d_k}})[V_s; V_t] \tag{3}$$

Since there are both self-attention ($Q_s K_s^\top$ and $Q_t K_t^\top$) for intra-domain feature fusion and cross-attention ($Q_s K_t^\top$ or $Q_t K_s^\top$) for inter-domain feature fusion, we call it the hybrid-attention module.

Furthermore, before aligning at the feature level, because the source and target domain data have different distributions in the feature space, their queries, keys, and values also vary in basis for calculating attention. When the key of the source and target domains are concatenated together, the network is forced to map the source and target domains on the same basis. In other words, for any query extracted by the network in any domain, it is completely equivalent to calculating the similarity under the basis of the source domain and the target domain. When the network learns this equivalence relationship, it achieves alignment between the two domains at the feature level.

Figure 2: **Illustration of the hybrid-attention mechanism and AAC.** Taking the feature tokens $f_s^i$, $f_t^i$ from source and target domains as inputs, three embedding layers project these tokens to the corresponding query $Q_i$, key $K_i$, and value tokens $V_i$ respectively, where $i \in \{s, t\}$. In hybrid-attention, query tokens $Q_t$ are from target domain, key $K_i$ and value tokens $V_i$ are formed by $[K_s; K_t]$ and $[V_s; V_t]$. After every stage of the Transformer(Xie et al., 2021), the AAC adaptively weights the sum of two features calculated from two attention mechanisms.

However, it is difficult for the network to directly learn unified feature representation ideally (Dai et al., 2021). Therefore, in the hybrid branch, a linear layer is used to perform quadratic mapping on the key and value of the source domain to the target domain.

$$Attn_{hybrid}(Q_t, K_s, K_t, V_s, V_t) = Softmax\left(\frac{Q_t[M_s^K(K_s); K_t]^\top}{\sqrt{d_k}}\right)[M_s^V(V_s); V_t] \qquad (4)$$

This way, even if the features of different domains cannot be fully aligned, the mapped query and key can be guaranteed to be on the same basis. Then we have the cross-entropy loss on the hybrid branch:

$$L_h^i = -\frac{1}{HW}\sum_{j=1}^{HW}\sum_{c=1}^{C}\hat{y}_t^{i,j,c}\log g^{(h)}(x_t^{(i)})^{(j,c)} \qquad (5)$$

where $g^{(h)}$ represents the forward function of the student Transformer network using hybrid-attention.

To further explore the feature alignment capability of the hybrid branch, in the following section, we introduce the proposed AAC, where dynamic alignment adjustment can be made based on specific semantic content.

## 3.2 ADAPTIVE ALIGNMENT CONTROLLER

Based on the current design, the model uses a domain query to query all keys in different domains, thereby calculating similarity masks and achieving feature alignment. However, We argue that aligning all the features at all stages is an oversight. For deep learning models, low-level layers extract features like color and texture, while high-level layers prefer semantic features. We hope to achieve adaptive alignment at different stages of the network. Even inside the high-level features, different layers should align different objects in different layers. Specifically, taking the GTA5 dataset and the Cityscapes dataset as examples, the road category of the two datasets has small difference in appearance, which is significant for vehicles. Therefore, we hope that the network should be able to align the road in relatively lower layers and align the vehicle in more higher layers where the features are more abstract focusing on semantic information rather than appearance information.

Aiming at learning domain informative representation at lower layers and domian uninformative representation at higher layers, CAN (Zhang et al., 2018) proposed a collaborative and adversarial learning scheme. However, the optimization weights learned by CAN can only be fixed for different network layers, and cannot consider fine-grained semantic information in semantic segmentation tasks. To solve this problem based on the observation above, we have designed the AAC behind the

hybrid-attention layer to adaptively adjust whether to update features using self-attention or hybrid-attention. As shown in Fig. 2, when training the hybrid branch, the network uses self-attention to calculate the feature $\hat{f}_{self}$, and use hybrid-attention to calculate the feature $\hat{f}_{hybr}$. Then a dense mask is learned for each feature map after every stage (Xie et al., 2021). The final feature map is combined by performing pixel-wise multiplication between masks with the two features.

More formally, the fused attention decoder $F^A$ learns to obtain the fused attention $a_f = \sigma(F^A(\hat{f}_{self})) \in [0,1]^{H \times W \times C}$ to weight the importance of hybrid feature. The sigmoid function $\sigma$ ensures a weight in $[0,1]$, where 0 means a focus on the origin domain, 1 means a focus on the hybrid domain. The final feature map is fused using the attention-weighted sum:

$$\hat{f} = a_f \odot \hat{f}_{self} + (1 - a_f) \odot \hat{f}_{hybr} \tag{6}$$

### 3.3 COORDINATE WEIGHT

Previous works mostly adopt a pre-trained model of the source domain as the initial model, and conduct an iterative process between pseudo-label generation and target domain training. Instead, our method jointly trains the images of the source and target domains end-to-end using three parallel branches. Therefore, it is crucial to coordinate the additional branch in the training process. Since no reliable pseudo-labels of the target domain are available at the beginning of the training, prematurely introducing hybrid attention brings label noise and prevents the model from learning discriminative semantic features. Conversely, introducing hybrid computation late can bias the model toward the source distribution and trap it in a local optimum.

Different from adjusting the weight of loss using the differences in network prediction results over time (Huang et al., 2021) or between different classifiers (Luo et al., 2019), we argue that the coordination between the source and target domain during training can be determined by the performance of the student model on the target domain. A well-performing student model means that the teacher model can provide reliable pseudo-labels. As a result, we propose to using the pseudo-accuracy of the target prediction $p_t$ (output by the student model) on the target pseudo-label $\hat{y}_t$ (output by the teacher model) as an anchor to control the participation of the target domain. The coordinate weight is defined as:

$$Coor(p_t, \hat{y}_t) = Acc(\bar{y}_t, \hat{y}_t) \times (1 - e^{-iter \cdot \alpha}), \quad \text{where } \bar{y}_t^{i,j} = \arg\max_k p_t^{i,j,k} \tag{7}$$

The iteration step is denoted as $iter$ and $\alpha$ is a hyperparameter to control the ascent speed. Therefore, the final loss is formulated as:

$$L = L_s + L_t + Coor(p_t, \hat{y}_t) \cdot L_h \tag{8}$$

We adopt HRDA as our teacher and student models. The B$^3$CT framework is taken as the baseline for the following experiments.

It is worth noting that we only use coordinate weight to control the loss value for the hybrid branch, and used the same weight for the segmentation cross-entropy loss of the source domain and target domain. The reason is that in the HRDA training framework, a quality/confidence estimate is produced for the pseudo labels. The ratio of pixels exceeding a threshold $T$ of the maximum softmax probability is used to calculate cross-entropy loss. However, in the hybrid domain, we calculate the hybrid attention uniformly for all tokens, so controlling coordinate weight on a time scale is a better choice to solve the "when to align" problem. Our method uses the accuracy of the student model on the target domain to dynamically determine the timing and the weight of engaging the target domain in training. Instead of adopting an exact threshold of accuracy to rigidly determine that the coordinate weight is either 0 or 1, using a smoothing varying coordinate weight can be taken as an advanced version and achieves better performance.

## 4 EXPERIMENTS

**Datasets:** We use GTA5 dataset (Richter et al., 2016) and Synthia dataset (Ros et al., 2016) for the source domain. GTA5 dataset consists of 24966 synthetic images with resolution 1914×1052. Synthia dataset contains 9,400 synthetic images with resolution 1280×760. For the target domain, we use the Cityscapes street scene dataset (Cordts et al., 2016), which contains 2975 training and 500 validation images with resolution 2048×2048.

Table 1: **Component Ablation for B$^3$CT**.

| hybrid branch | coor.weight | AAC | mIoU |
|:---:|:---:|:---:|:---:|
| – | – | – | 73.8 |
| ✓ | – | – | 74.2 |
| ✓ | ✓ | – | 74.3 |
| ✓ | ✓ | ✓ | **74.8** |

Table 2: **Different data flow in hybrid branch**.

| source | target | mIoU |
|:---:|:---:|:---:|
| – | – | 73.8 |
| ✓ | – | 73.1 |
| – | ✓ | **74.8** |
| ✓ | ✓ | 74.5 |

**Implementation Details:** We use HRDA (Hoyer et al., 2022b) as our default network, which consists of an MIT-B5 encoder (Xie et al., 2021), a context-aware feature fusion decoder (Hoyer et al., 2022a) and a lightweight SegFormer MLP decoder (Xie et al., 2021) as the scale attention decoder. The self-training strategy and training parameters are all the same as HRDA's settings. We use AdamW with a learning rate of $6 \times 10^{-5}$ for the encoder and $6 \times 10^{-4}$ for the decoder, linear learning rate warmup, and DACS (Tranheden et al., 2021) data augmentation. The batch size is set as 2. The experiments are conducted on a Titan RTX GPU with 24GB memory using mixed precision training. All of the results are averaged over 3 random seeds.

## 4.1 Ablation studies for B$^3$CT

**Component ablation for B$^3$CT.** In our B$^3$CT self-training framework, the main contributions can be divided into three parts: additional hybrid branch, AAC for selective feature alignment, and the use of coordinate weight to gradually enhance the participation of hybrid branches. The components are ablated in Tab. 1. Three parts were added in sequence, resulting in improvements of mIoU 0.37, 0.16, and 0.42, respectively. It can be seen that the main performance improvement comes from the feature alignment of the third branch, while the coordinate weight and AAC module have added the icing on the cake.

Table 3: **Ablation study of the hybrid-attention with AAC in each stage.** The first line shows the performance without the third branch.

| Stage1 | Stage2 | Stage3 | Stage4 | mIoU |
|:---:|:---:|:---:|:---:|:---:|
| – | – | – | – | 73.8 |
| ✓ | – | – | – | 74.3 |
| – | ✓ | – | – | 74.3 |
| – | – | ✓ | – | 74.4 |
| – | – | – | ✓ | 74.4 |
| ✓ | ✓ | – | – | 74.3 |
| – | ✓ | ✓ | – | 74.5 |
| – | – | ✓ | ✓ | 74.6 |
| ✓ | ✓ | ✓ | – | 74.5 |
| – | ✓ | ✓ | ✓ | 74.6 |
| ✓ | ✓ | ✓ | ✓ | **74.8** |

**Different data flow in the hybrid branch.** Our B$^3$CT framework has three parallel branches, namely the source branch, target branch, and hybrid branch. Among them, the source branch and target branch are respectively calculated by self-attention from source and target images to obtain prediction and perform backpropagation. In the hybrid branch, we tested three different data flows. When the data flow is only the source image, the hybrid-attention layer and AAC provide query by the source image, and the images of the two domains jointly provide the key and value. Finally, only the prediction of the source domain is calculated and backpropagated in the hybrid branch. When the data flow is source + target images, the hybrid-attention layer and AAC obtain two feature maps based on the two domain queries, ultimately obtaining source and target predictions together.

Experiments in Tab. 2 have shown that the model achieves optimal results when only target data is used as the third branch. It indicates that aligning source to target is better than aligning target to source, which is consistent with our task setting, i.e., a good model on the target domain. Moreover, the dataflow of source+target is also better than the dataflow with only source data. It further implies that the proposed hybrid branch works well on feature alignment.

**Hybrid-attentionin different stages.** In order to verify that the alignment of each stage is necessary, in Tab. 3, we demonstrate the effect of hybrid-attention in different Transformer stages. Since the alignment processes in the four stages increase, the adaptation performance of our method arises from 73.8 to 74.8. The experimental results indicate that aligning features in each layer of the network makes sense. More hybrid-attention are applied, the tighter the fusion of features between

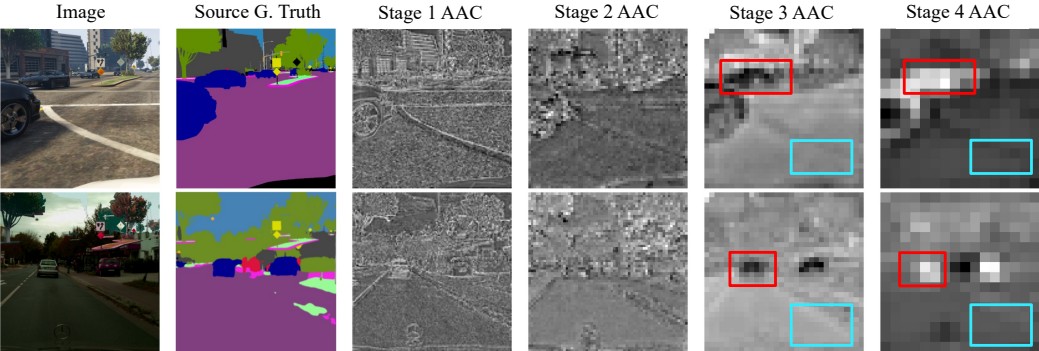

Figure 3: **Visual examples of the AAC in different Transformer(Xie et al., 2021) stage.** The brighter the color, the greater the feature weight calculated using hybrid-attention, while the darker the color, the greater the feature weight calculated using self-attention. The DAFI learns to decide whether features from two domains should be aligned at certain stages.

the source and target domains, the more it helps the model learn domain-invariant features. It is worth noting that in this experiment, all the hybrid-attentions are together with AAC.

## 4.2 ABLATION STUDIES FOR THE AAC

**Quantitative experiments on AAC.** More specifically on the AAC module, we conduct quantitative experimental verification on this module in Tab. 4. If only the hybrid-attention is conducted without adding AAC, the final model mIoU is 74.3. Given that the function implemented by AAC is the dynamic weighting of two feature maps, we attempt to directly average the two feature maps and result in no improvement. However, Applying AAC will result in an improvement of 0.42 mIoU. It can be seen that directly using hybrid-attention for feature alignment can easily lead to suboptimal results due to domain gaps. Using AAC to adaptively align and adjust each token at each stage can achieve the optimal alignment effect.

Table 4: **Ablation study for AAC**. The AAC is compared with no feature fusion strategy and simple average pooling.

| Feature fusion | mIoU |
| --- | --- |
| – | 74.3 |
| Average Pooling | 74.4 |
| Fused Attention | **74.8** |

**Qualitive experiments on AAC.** Furthermore, in Fig. 3, we visualized the alignment strategies learned by the AAC module at each stage of the network. In the figure, brighter colored tokens represent AAC giving greater weight to the features calculated by hybrid-attention; On the contrary, darker tokens represent AAC focusing more on self-attention features. In other words, a brighter color indicates that the network is more inclined to align the feature representations of the token across different domains certain stages.

Firstly, from stage 1, it can be seen that the network mainly aligns at texture and edges. This pattern is similar to using CycleGAN (Zhu et al., 2017) to align images from different domains at the input level, such as making categories from different domains have the same texture and color information. As the network layer deepens, AAC increasingly tends to align at the level of semantic object features. For example, in Fig. 3, AAC aligns the road and other categories in stage 3, emphasized by the blue boxes, and aligns the vehicle and other categories in stage 4, emphasized by the red boxes. These results are consistent with our motivation for designing AAC.

## 4.3 ABLATION STUDIES FOR COORDINATE WEIGHT

To explore the most suitable feature alignment time, we investigate the effects of the coordinate weight. The hyperparameter $\alpha$ of coordinate weight in Eq. 7 is designed to achieve harmonious training among the three branches. We plot the curves of coordinate weights with different $\alpha$ on the left of Fig. 4. The corresponding performance is shown on the right of Fig. 4. Setting $\alpha = 0$ means that the hybrid branch is not involved and only the source branch and target branch are available for

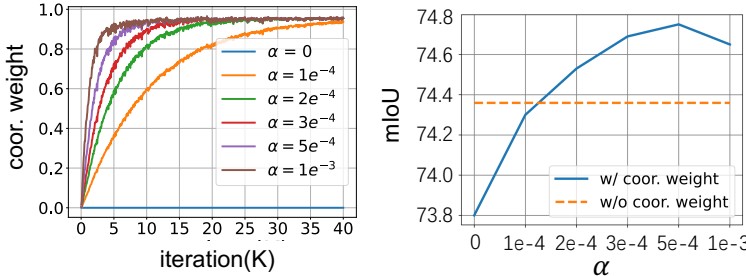

Figure 4: Coordinate weight curves and performance comparison between various $\alpha$. Setting $\alpha = 0$ indicates that the hybrid branch is not involved in the training. Our method reaches 74.4 without coordinate weight as shown in the dashed line.

Table 5: Comparison with exsiting methods on GTAV/Synthia to Cityscapes benchmarks. The **first** and second highest scores are represented by bold font and underline respectively.

| | Road | S.walk | Build. | Wall | Fence | Pole | Tr.Light | Sign | Veget. | Terrain | Sky | Person | Rider | Car | Truck | Bus | Train | M.bike | Bike | mIoU |
|---|---|---|---|---|---|---|---|---|---|---|---|---|---|---|---|---|---|---|---|---|
| | | | | | | | | GTA5 → Cityscapes | | | | | | | | | | | | |
| CBST (Zou et al., 2018) | 91.8 | 53.5 | 80.5 | 32.7 | 21.0 | 34.0 | 28.9 | 20.4 | 83.9 | 34.2 | 80.9 | 53.1 | 24.0 | 82.7 | 30.3 | 35.9 | 16.0 | 25.9 | 42.8 | 45.9 |
| DACS (Tranheden et al., 2021) | 89.9 | 39.7 | 87.9 | 30.7 | 39.5 | 38.5 | 46.4 | 52.8 | 88.0 | 44.0 | 88.8 | 67.2 | 35.8 | 84.5 | 45.7 | 50.2 | 0.0 | 27.3 | 34.0 | 52.1 |
| DPL-Dual (Cheng et al., 2021) | 92.8 | 54.4 | 86.2 | 41.6 | 32.7 | 36.4 | 49.0 | 34.0 | 85.8 | 41.3 | 86.0 | 63.2 | 34.2 | 87.2 | 39.3 | 44.5 | 18.7 | 42.6 | 43.1 | 53.3 |
| SAC (Araslanov & Roth, 2021) | 90.4 | 53.9 | 86.6 | 42.4 | 27.3 | 45.1 | 48.5 | 42.7 | 87.4 | 40.1 | 86.1 | 67.5 | 29.7 | 88.5 | 49.1 | 54.6 | 9.8 | 26.6 | 45.3 | 53.8 |
| CorDA (Wang et al., 2021a) | 94.7 | 63.1 | 87.6 | 30.7 | 40.6 | 40.2 | 47.8 | 51.6 | 87.6 | 47.0 | 89.7 | 66.7 | 35.9 | 90.2 | 48.9 | 57.5 | 0.0 | 39.8 | 56.0 | 56.6 |
| ProDA (Zhang et al., 2021) | 87.8 | 56.0 | 79.7 | 46.3 | 44.8 | 45.6 | 53.5 | 53.5 | 88.6 | 45.2 | 82.1 | 70.7 | 39.2 | 88.8 | 45.5 | 59.4 | 1.0 | 48.9 | 56.4 | 57.5 |
| DAFormer (Hoyer et al., 2022a) | 95.7 | 70.2 | 89.4 | 53.5 | 48.1 | 49.6 | 55.8 | 59.4 | 89.9 | 47.9 | 92.5 | 72.2 | 44.7 | 92.3 | 74.5 | 78.2 | 65.1 | 55.9 | 61.8 | 68.3 |
| HRDA (Hoyer et al., 2022b) | 96.4 | 74.4 | 91.0 | **61.6** | 51.5 | 57.1 | 63.9 | **69.3** | 91.3 | 48.4 | 94.2 | 79.0 | 52.9 | 93.9 | **84.1** | 85.7 | 75.9 | **63.9** | 67.5 | 73.8 |
| DAFormer + B³CT | **96.5** | **75.2** | 90.7 | 60.6 | 45.5 | 57.5 | 63.7 | 68.6 | 90.9 | 48.4 | 91.9 | 76.7 | 49.2 | 92.1 | 59.5 | 68.3 | 64.6 | 58.7 | 67.9 | 69.8 |
| HRDA + B³CT | **96.5** | 75.0 | **91.2** | 59.6 | **55.4** | **58.5** | **66.3** | 69.1 | **91.9** | **49.3** | **94.4** | **79.2** | **55.3** | **94.2** | 83.2 | **88.7** | **80.3** | 63.7 | **68.4** | **74.8** |
| | | | | | | | | Synthia → Cityscapes | | | | | | | | | | | | |
| CBST (Zou et al., 2018) | 68.0 | 29.9 | 76.3 | 10.8 | 1.4 | 33.9 | 22.8 | 29.5 | 77.6 | – | 78.3 | 60.6 | 28.3 | 81.6 | – | 23.5 | – | 18.8 | 39.8 | 42.6 |
| DACS (Tranheden et al., 2021) | 80.6 | 25.1 | 81.9 | 21.5 | 2.9 | 37.2 | 22.7 | 24.0 | 83.7 | – | 90.8 | 67.6 | 38.3 | 82.9 | – | 38.9 | – | 28.5 | 47.6 | 48.3 |
| DPL-Dual (Cheng et al., 2021) | 83.5 | 38.2 | 80.4 | 1.3 | 1.1 | 29.1 | 20.2 | 32.7 | 81.8 | – | 83.6 | 55.9 | 20.3 | 79.4 | – | 26.6 | – | 7.4 | 46.2 | 43.0 |
| SAC (Araslanov & Roth, 2021) | 89.3 | 47.2 | 85.5 | 26.5 | 1.3 | 43.0 | 45.5 | 32.0 | 87.1 | – | 89.3 | 63.6 | 25.4 | 86.9 | – | 35.6 | – | 30.4 | 53.0 | 52.6 |
| CorDA (Wang et al., 2021a) | **93.3** | **61.6** | 85.3 | 19.6 | 5.1 | 37.8 | 36.6 | 42.8 | 84.9 | – | 90.4 | 69.7 | 41.8 | 85.6 | – | 38.4 | – | 32.6 | 53.9 | 55.0 |
| ProDA (Zhang et al., 2021) | 87.8 | 45.7 | 84.6 | 37.1 | 0.6 | 44.0 | 54.6 | 37.0 | 88.1 | – | 84.4 | 74.2 | 24.3 | 88.2 | – | 51.1 | – | 40.5 | 45.6 | 55.5 |
| DAFormer (Hoyer et al., 2022a) | 84.5 | 40.7 | 88.4 | 41.5 | 6.5 | 50.0 | 55.0 | 54.6 | 86.0 | – | 89.8 | 73.2 | 48.2 | 87.2 | – | 53.2 | – | 53.9 | 61.7 | 60.9 |
| HRDA (Hoyer et al., 2022b) | 85.2 | 47.7 | **88.8** | **49.5** | 4.8 | 57.2 | 65.7 | 60.9 | 85.3 | – | 92.9 | 79.4 | 52.8 | **89.0** | – | **64.7** | – | 63.9 | 64.9 | 65.8 |
| DAFormer + B³CT | 82.8 | 44.2 | 86.7 | 38.9 | 5.0 | 55.0 | 63.0 | 61.2 | 83.6 | – | 82.9 | 76.3 | 47.9 | 88.0 | – | 58.6 | – | 61.8 | 65.3 | 62.6 |
| HRDA + B³CT | 90.5 | 57.5 | 88.7 | 46.3 | **7.5** | **57.7** | **66.6** | **63.3** | **89.1** | – | **94.2** | **80.5** | **55.8** | **89.0** | – | 54.4 | – | **64.6** | **65.7** | **67.0** |

model training. As the $\alpha$ increases, the earlier the target domain is involved in the training, while the overall model effect shows a trend of first increasing and then decreasing. We take $\alpha = 5e^{-4}$ as our default setting according to the experimental results.

### 4.4 COMPARISON WITH EXISTING METHODS

To show the superiority of the proposed method, we report adaptation performance in terms of mIoU (%) on two benchmarks in Tab. 5. B³CT provides a 1.5 and 1.0 percent point improvement over the respective DAFormer and HRDA performance on GTAV-to-Cityscapes, 1.7 and 1.2 percent point on Synthia-to-Cityscapes. This implies that B³CT has the flexibility to seamlessly integrate with various Transformer-based models. Additionally, B³CT also outperforms the baseline on the Citsycapes-to-ACDC benchmark. Detailed performance can be found in the appendix.

## 5 CONCLUSION

In this work, we focus on achieving efficient alignment of features between different domains in domain adaptive semantic segmentation tasks and raise two basic issues: "where to align" and "when to align". To solve these two issues comprehensively, we first propose a Three-Branch Coordinated Training (B³CT) framework. In B³CT, its third branch intuitively consists of intra-domain self-attention and inter-domain cross-attention to achieve feature fusion and alignment, where an Adaptive Alignment Controller (AAC) is designed to conduct alignment at the right stage for the right contents. Moreover, we propose a coordinate weight to dynamically adjust the alignment time throughout the entire training process. Our method solves the problem of coordination between the learning of discriminative category features from the source domain and the learning of feature distribution of the target domain. We have shown that our method achieves competitive performance in both GTA5→Cityscapes and SYNTHIA→Cityscapes benchmarks.

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

# A APPENDIX

## A.1 COMPARISON ON FURTHER BENCHMARK

Table 6: **Comparison with baselines on Cityscapes to ACDC benchmark.** The results are obtained on the held-out test set of ACDC whose annotation is not accessible. * represents results from our re-implementation. † represents methods use additional clear-weather geographically-aligned reference images. The **first** and second highest scores are represented by bold font and underline respectively.

| | Road | S.walk | Build. | Wall | Fence | Pole | Tr.Light | Sign | Veget. | Terrain | Sky | Person | Rider | Car | Truck | Bus | Train | M.bike | Bike | mIoU |
|---|---|---|---|---|---|---|---|---|---|---|---|---|---|---|---|---|---|---|---|---|
| | | | | | | | Cityscapes → ACDC | | | | | | | | | | | | | |
| ADVENT (Vu et al., 2019) | 72.9 | 14.3 | 40.5 | 16.6 | 21.2 | 9.3 | 17.4 | 21.2 | 63.8 | 23.8 | 18.3 | 32.6 | 19.5 | 69.5 | 36.2 | 34.5 | 46.2 | 26.9 | 36.1 | 32.7 |
| MGCDA† (Sakaridis et al., 2020) | 73.4 | 28.7 | 69.9 | 19.3 | 26.3 | 36.8 | 53.0 | 53.3 | **75.4** | 32.0 | 84.6 | 51.0 | 26.1 | 77.6 | 43.2 | 45.9 | 53.9 | 32.7 | 41.5 | 48.7 |
| DANNet† (Wu et al., 2021) | 84.3 | 54.2 | 77.6 | 38.0 | 30.0 | 18.9 | 41.6 | 35.2 | 71.3 | 39.4 | **86.6** | 48.7 | 29.2 | 76.2 | 41.6 | 43.0 | 58.6 | 32.6 | 43.9 | 50.0 |
| DAFormer (Hoyer et al., 2022a) | 58.4 | 51.3 | 84.0 | 42.7 | 35.1 | 50.7 | 30.0 | 57.0 | 74.8 | 52.8 | 51.3 | 58.3 | 32.6 | 82.7 | 58.3 | 54.9 | 82.4 | 44.1 | 50.7 | 55.4 |
| HRDA* (Hoyer et al., 2022b) | 69.1 | **64.8** | 86.4 | **53.9** | 40.3 | **56.5** | 41.1 | **63.9** | 72.8 | **58.8** | 62.0 | **69.9** | 44.2 | 87.0 | **73.5** | 82.0 | 87.8 | 45.4 | **63.0** | 64.3 |
| HRDA* + B³CT | **89.8** | 62.0 | **86.6** | 53.5 | **40.7** | 48.9 | **58.8** | 60.1 | 73.2 | 57.1 | 85.7 | 67.9 | **44.6** | **87.9** | 72.5 | 84.3 | **88.4** | **51.6** | 56.8 | **66.9** |

In the main text, both GTA5 → Cityscapes and Synthia → Cityscapes represent domain adaptation tasks from synthetic scenes to real-world environments. In order to assess the generalization capabilities of our proposed method across diverse scenarios, we conducted additional validation on the clear-to-adverse-weather Cityscapes → ACDC benchmark, which involves domain adaptation from clear weather conditions in Cityscapes to adverse weather conditions. The results are shown in 6. In comparison to our reimplementation of HRDA achieving a mIoU of 64.3 on the test set, B³CT further enhances performance to 66.9 mIoU. These experimental results provide further evidence that our approach can achieve alignment across multiple domains, demonstrating superior performance in the target domain.

## A.2 COMPARISON OF COMPUTATIONAL EFFICIENCY

Table 7: **Comparison of runtime and parameters.** Runtime and parameters of different UDA methods on an Nvidia Titan RTX.

| UDA method | Throughput(img/s) | TFLOPs | Training params. |
|---|---|---|---|
| DAFormer | 0.7 | 1.6 | 85.2M |
| HRDA | 0.8 | 3.3 | 85.7M |
| HRDA* + B³CT | 0.8 | 3.3 | 86.1M |

The runtime and parameters of HRDA during inference is shown in 7. Our proposed method is compared against DAFormer and HRDA across key runtime metrics. Notably, the hybrid branch is only added during the training phrase. It has no adverse impact on inference speed during testing. Therefore, our model has the same throughput (0.8) and TFLOPs (3.3) as HRDA. Despite of the runtime, our approach maintains minimal computational burden, with only a marginal increase in parameters (86.1M) compared to DAFormer (85.2M) and HRDA (85.7M). In summary, our method showcases competitive performance without compromising computational efficiency or inference speed.

