# OpenReview forum: "B$^{3}$CT: Three-branch Coordinated Training for Domain Adaptive Semantic Segmentation"
_ICLR.cc/2024/Conference — Submitted to ICLR 2024_

### Official Review · Reviewer_Kyed · 2023-10-30

**Soundness:** 2 fair
**Presentation:** 2 fair
**Contribution:** 2 fair
**Rating:** 3
**Confidence:** 5

**Summary:**

This work aims to address the problem of domain adaptive semantic segmentation, mainly focusing on "where and when to align". At first, a hybrid-attention mechanism is proposed to achieve feature fusion and alignment. Then, an adaptive alignment controller (AAC) is designed to determine the alignment feature at each stage. Next, a coordinate weight is proposed to adjust the alignment time through the training process. In summary, the main contributions of this work are somehow novel, however, the performance gains are limited, and the comparison results are insufficient.

**Strengths:**

**Originality**: The paper proposes three ideas, hybrid branch, adaptive alignment controller (AAC), and coordinate weight, which are generally applicable and work for feature alignment and fusion.

**Quality**: The paper provides a thorough experimental evaluation of B$^3$CT on two simulation-to-real domain adaptive semantic segmentation bechmarks. The paper also conducts ablation studies to analyze the impact of different components of B$^3$CT, such as data flow in the hybrid branch, qualitative experiments on AAC, and the hyperparameter of coordinate weight. The paper demonstrates that B$^3$CT can achieve superior performance when combined with the HRDA baseline.

**Clarity**: The paper also provides sufficient background information and related work to situate the contribution of B$^3$CT in the context of existing literature on domain adaptive semantic segmentation and self- and cross-attention.

**Significance**: The paper addresses an important and challenging problem of domain adaptive semantic segmentation, which has many applications in autonomous driving, robotics, and scene understanding.

**Weaknesses:**

**Major Issues**:

**Insufficient novelty and contribution**: The newly proposed B$^3$CT framework lacks justification for its design. The pipelines of hybrid-attention and adaptive alignment controller seem natural and basic.

**Insufficient results for experiments**:

- Although the authors state in the main text, "the third branch of which facilitates learning domain-invariant features for alignment", they provide no experimental results.

- In Tab. 1, the result of only ablating "coor. weight" also should be reported.

- In Sec. 4.4, the authors should differentiate the comparison results according to different network architectures, such as deeplab and segformer.

- Recent works not only conduct experiments on two standard simulation-to-real benchmarks, i.e., GTA5-to-Cityscapes and SYNTHIA-to-Cityscapes but also extend to adverse conditions. To name a few, SePiCo [a] and MIC [b] have extended to more challenging daytime-to-nighttime semantic segmentation task and CoTTA [c] has also investigated clear-to-adverse conditions using online adaptation. I would like to see the potential of B$^3$CT on more challenging scenarios.

**Insufficient justifications**: For example, about training accuracy or pseudo-accuracy, some justifications are missing in this paper. Any advantages and limitations of pseudo-accuracy?

**Insufficient details**: From Tab. 3, the only thing we can see is that there will be one hybrid attention with AAC in each stage. However, where should we exactly insert hybrid branch into a feature encoder? And what is the consumption of resources, such as GPU memory?

**Minor Issues**:

- In Eq. 2 and Eq. 5, $p_t^{i,j,c}$ (target predictions from student model) should be $\hat{y}_t^{i,j}$ (target pseudo-label from teacher model)? Also, in Eq. 7 and Eq. 8 $p_t^i, \hat{y}_t^i$ should be $p_t^{i,j}, \hat{y}_t^{i,j}$, respectively?

- The results of the experiments throughout the text are in two retained decimals, except for Table 5, where the authors should be consistent.

- Typos: "hybrid-attention different stages." -> "Hybrid-attention different stages." in Sec. 4.1.


Refs:

[a] Xie et al. SePiCo: Semantic-Guided Pixel Contrast for Domain Adaptive Semantic Segmentation, TPAMI 2023.

[b] Hoyer et al. MIC: Masked Image Consistency for Context-Enhanced Domain Adaptation, CVPR 2023.

[c] Wang et al. Continual Test-Time Domain Adaptation, CVPR 2022.

**Questions:**

The authors should discuss the limitations and potential negative societal impact in the Conclusion.

Please also refer to Weaknesses.

---

> ### Author Response · Authors · 2023-11-22
> **Response [1/3]**
>
> We thank the reviewer for the constructive feedback and provide responses below.
>
> **Q1: Experimental results of the third branch.**
>
> **A1:** As you mentioned, ablation experiments on the third branch are very necessary. Our method is a three-branch collaborative training method. If only the third branch that uses hybrid attention for calculation is added, the performance will also be improved. We have included this experiment in Tab. 1 of the paper. The table is also demonstrated below. Adding only the third branch can bring an improvement of 0.4 mIoU. The improvement effect is not significant because this branch needs to collaborate with other modules. Only by adding all three components can the best effect be achieved.
>
> #### Component Ablation for B3CT
>
> | hybrid branch | coor.weight | AAC |   mIoU   |
> |-|-|-|-|
> | - | - |  -  |   73.8   |
> | ✓ | - |  -  |   74.2   |
> | ✓ | ✓ |  -  |   74.3   |
> | ✓ | ✓ |  ✓  | **74.8** |
>
> **Q2: Results of only ablating coordinate weight.**
>
> **A2:** We apologize for any lack of clarity in our initial description. In our B3CT framework, coordinate weight is employed to dynamically adjust the contribution of the third branch during model training. Therefore, applying coordinate weight necessitates the inclusion of the third branch. Additionally, we conducted ablation experiments with different hyperparameters for the coordinate weight $\alpha$, and the specific results are shown in Figure 4.
>
> **Q3: The comparison results of different network architectures.**
>
> **A3:** Investigating the generalization ability of a model is crucial. We applied the B3CT model using DAFormer as the backbone on two benchmarks, GTA5 $\rightarrow$ Cityscapes and Synthia $\rightarrow$ Cityscapes respectively. The experimental results are shown in the table below and updated in Tab. 5. B3CT achieves a 1.5 and 1.7 percent point improvement over the respective two benchmarks. This implies that B3CT has the flexibility to integrate with various Transformer-based models seamlessly.
>
> #### Further baseline comparison on GTA5 $\rightarrow$ Cityscapes
> || Road | S.walk | Build. | Wall | Fence | Pole | Tr.Light | Sign | Veget. | Terrain | Sky | Person | Rider | Car | Truck | Bus | Train | M.bike | Bike | mIoU |
> |-|-|-|-|-|-|-|-|-|-|-|-|-|-|-|-|-|-|-|-|-|
> | DAFormer    | 95.7 | 70.2 | 89.4 | 53.5 | 48.1 | 49.6 | 55.8 | 59.4 | 89.9 | 47.9 | 92.5 | 72.2 | 44.7 | 92.3 | 74.5 | 78.2 | 65.1 | 55.9 | 61.8 | 68.3 |
> | HRDA | 96.4 | 74.4 | 91.0 | **61.6**    | 51.5 | 57.1 | 63.9 | **69.3**    | 91.3 | 48.4 | 94.2 | 79.0 | 52.9 | 93.9 | **84.1**    | 85.7 | 75.9 | **63.9**    | 67.5 | 73.8 |
> | DAFormer + B3CT | **96.5**    | **75.2**    | 90.7 | 60.6 | 45.5 | 57.5 | 63.7 | 68.6 | 90.9 | 48.4 | 91.9 | 76.7 | 49.2 | 92.1 | 59.5 | 68.3 | 64.6 | 58.7 | 67.9 | 69.8 |
> | HRDA + B3CT | **96.5**    | 75.0 | **91.2**    | 59.6 | **55.4**    | **58.5**    | **66.3**    | 69.1 | **91.9**    | **49.3**    | **94.4**    | **79.2**    | **55.3**    | **94.2**    | 83.2 | **88.7**    | **80.3**    | 63.7 | **68.4**    | **74.8**   |
>
> #### Further baseline comparison on Synthia $\rightarrow$ Cityscapes
> || Road | S.walk | Build. | Wall | Fence | Pole | Tr.Light | Sign | Veget. | Terrain | Sky | Person | Rider | Car | Truck | Bus | Train | M.bike | Bike | mIoU |
> |-|-|-|-|-|-|-|-|-|-|-|-|-|-|-|-|-|-|-|-|-|
> | DAFormer    | 84.5 | 40.7 | 88.4 | 41.5 | 6.5  | 50.0 | 55.0 | 54.6 | 86.0 | -- | 89.8 | 73.2 | 48.2 | 87.2 | -- | 53.2 | -- | 53.9 | 61.7 | 60.9 |
> | HRDA | 85.2 | 47.7 | **88.8**    | **49.5**    | 4.8 | 57.2 | 65.7 | 60.9 | 85.3 | -- | 92.9 | 79.4 | 52.8 | **89.0**    | -- | **64.7**    | -- | 63.9 | 64.9 | 65.8 |
> | DAFormer + B3CT | 82.8 | 44.2 | 86.7 | 38.9 | 5.0 | 55.0 | 63.0 | 61.2 | 83.6 | -- | 82.9 | 76.3 | 47.9 | 88.0 | -- | 58.6 | -- | 61.8 | 65.3 | 62.6 |
> | HRDA + B3CT | 90.5 | 57.5 | 88.7 | 46.3 | **7.5**     | **57.7**    | **66.6**    | **63.3**    | **89.1**    | -- | **94.2**    | **80.5**    | **55.8**    | **89.0**    | -- | 54.4 | -- | **64.6**    | **65.7**    | **67.0**  |

---

> > ### Author Response · Authors · 2023-11-22
> > **Response [2/3]**
> >
> > **Q4: Experiments on more challenging scenarios.**
> >
> > **A4:** To assess the generalization capabilities of our proposed method across diverse scenarios, we conducted additional validation on the clear-to-adverse-weather Cityscapes $\to$ ACDC benchmark, which involves domain adaptation from clear weather conditions in Cityscapes to adverse weather conditions. Our re-implementation of HRDA achieves 64.3 mIoU on the test set, which is basically consistent with third-party re-implementation results. In comparison to this, B3CT further enhances performance to 66.9 mIoU. These experimental results prove the robustness of our B3CT. More detailed experimental results can be found in the table below and App. A.1.
> >
> > #### Comparison with existing methods on Cityscapes $\rightarrow$ ACDC
> >
> > || Road | S.walk | Build. | Wall | Fence | Pole | Tr.Light | Sign | Veget. | Terrain | Sky | Person | Rider | Car | Truck | Bus | Train | M.bike | Bike | mIoU |
> > |----|-|-|-|-|-|-|-|-|-|-|-|-|-|-|-|-|-|-|-|-|
> > | ADVENT | 72.9 | 14.3 | 40.5 | 16.6 | 21.2 | 9.3 | 17.4 | 21.2 | 63.8 | 23.8 | 18.3 | 32.6 | 19.5 | 69.5 | 36.2 | 34.5 | 46.2 | 26.9 | 36.1 | 32.7 |
> > | MGCDA | 73.4 | 28.7 | 69.9 | 19.3 | 26.3 | 36.8 | 53.0 | 53.3 | **75.4**    | 32.0 | 84.6 | 51.0 | 26.1 | 77.6 | 43.2 | 45.9 | 53.9 | 32.7 | 41.5 | 48.7 |
> > | DANNet    | 84.3 | 54.2 | 77.6 | 38.0 | 30.0 | 18.9 | 41.6 | 35.2 | 71.3 | 39.4 | **86.6**    | 48.7 | 29.2 | 76.2 | 41.6 | 43.0 | 58.6 | 32.6 | 43.9 | 50.0 |
> > | DAFormer | 58.4 | 51.3 | 84.0 | 42.7 | 35.1 | 50.7 | 30.0 | 57.0 | 74.8 | 52.8 | 51.3 | 58.3 | 32.6 | 82.7 | 58.3 | 54.9 | 82.4 | 44.1 | 50.7 | 55.4 |
> > | HRDA | 69.1 | **64.8** | 86.4 | **53.9**    | 40.3 | **56.5**    | 41.1 | **63.9**    | 72.8 | **58.8**    | 62.0 | **69.9**    | 44.2 | 87.0 | **73.5**    | 82.0 | 87.8 | 45.4 | **63.0**    | 64.3 |
> > | HRDA + B3CT | **89.8**    | 62.0 | **86.6**    | 53.5 | **40.7**    | 48.9 | **58.8**    | 60.1 | 73.2 | 57.1 | 85.7 | 67.9 | **44.6**    | **87.9**    | 72.5 | **84.3**    | **88.4**    | **51.6**    | 56.8 | **66.9**    |
> >
> > **Q5: Justifications about pseudo-accuracy.**
> >
> > **A5:** Our model employs a self-training strategy, where the teacher model generates pseudo-labels to supervise the student model during training. Due to the absence of ground truth in the target domain, assessing the accuracy of pseudo-labels becomes challenging. If the accuracy of pseudo-labels is compromised, enforcing domain alignment on the third branch may lead to network oscillation and convergence issues. We extensively discuss this matter in Sec. 3.3, with the original text citation:
> >
> >     Since no reliable pseudo-labels of the target domain are available at the beginning of the training, prematurely introducing hybrid attention brings label noise and prevents the model from learning discriminative semantic features. Conversely, introducing hybrid computation late can bias the model toward the source distribution and trap it in a local optimum.
> >
> > In summary, using pseudo-labels as a factor for coordinate weight weights is one of its advantages. The possible limitation is that there may be errors due to distribution bias in pseudo-labels, but the outputs of the student model and the teacher model are almost identical. This is an unavoidable issue for UDA tasks because the target domain does not have any ground truth. We will continue to explore this direction in the future to see if there is a more suitable coordinate weight design.
> >
> > **Q6: Insufficient details.**
> >
> > **A6:** As you pointed out, the AAC module is implemented after each network stage. For models with HRDA as the backbone, four AAC modules are incorporated. A hybrid branch traverses the entire network, introducing three data streams in each stage: source branch, target branch, and hybrid branch. The AAC module dynamically determines the extent to which target branch output influences the hybrid branch output, governing the degree of domain alignment. Detailed explanations have been added in Sec. 3.1.
> >
> > In the Appendix, we provide the runtime and parameters of HRDA during inference. **Firstly**, the hybrid branch is only added during the training phase, with no adverse impact on inference speed during testing. Consequently, our model maintains the same throughput (0.8) and TFLOPs (3.3) as HRDA. **Secondly**, our model introduces a marginal increase of 0.4M parameters during the training phase, negligible compared to the overall parameter count (86.1M). In summary, our method achieves competitive performance while ensuring computational efficiency and rapid inference speed.
> >
> > #### Comparison of computational efficiency
> >
> > | UDA method  | Throughput(img/s) | TFLOPS | Training Params. |
> > |-|-|-|-|
> > | DAFormer    | 0.7 | 1.6    | 85.2M   |
> > | HRDA | 0.8 | 3.3    | 85.7M   |
> > | HRDA + B3CT | 0.8 | 3.3    | 86.1M   |

---

> > > ### Author Response · Authors · 2023-11-22
> > > **Response [3/3]**
> > >
> > > **Q7: Other minor issues.**
> > >
> > > **A7:** Thank you for your meticulous review. We have addressed the mentioned issues accordingly:
> > > - Corrections have been made to the symbols in the equations as identified.
> > >
> > > - To facilitate better comparison with previous work, all experimental data have been adjusted to retain one decimal point, ensuring consistency in result formatting.
> > >
> > > - Capitalization errors in Section 4.1 have been rectified.
> > >
> > > Thank you for your thoughtful comments and suggestions again.

---

> > > > ### Comment · Reviewer_Kyed · 2023-11-22
> > > > **Thank you for your detailed rebuttals.**
> > > >
> > > > I appreciate the results of Cityscapes -> ACDC provided by the authors. However, I am still not convinced on the following aspects:
> > > >
> > > > first, there is no response about **Insufficient novelty and contribution**.
> > > >
> > > > second, as for the main results present in the preliminary manuscript, all reviewers (i.e., qoMd, YWN9, Prz3, Kyed) agree that the performance gain is limited. For me, the authors should not compare baselines that use DAFormer as the backbone with those of DeepLab without any explanation. Further, there is no point in providing the results in the original text again.
> > > >
> > > > a warm reminder, all references mentioned by reviewers should be discussed and compared as much as possible.
> > > >
> > > >
> > > > overall, either one of the issues is fine (i.e. if the method performs exceptionally well but lacks clear motivations, or the method has clear intuition but performance might not be great across the board), the combination of the two leads me to keep my rating at reject.

---

### Official Review · Reviewer_Prz3 · 2023-10-30

**Soundness:** 3 good
**Presentation:** 3 good
**Contribution:** 2 fair
**Rating:** 5
**Confidence:** 5

**Summary:**

The paper primarily focuses on the domain-adaptive semantic segmentation task. The authors introduce a so-called "Three-Branch Coordinated Training" (B^3CT) framework. This framework encompasses distinct source and target domain branches, along with a mixed attention branch equipped with an Alignment Controller (AAC) to transfer knowledge from the source to the target domain gradually. Additionally, the authors propose a "coordinate weight" strategy to emphasize when to execute the knowledge transfer. The authors have effectively evaluated their approach to major public benchmarks.

**Strengths:**

1. The problem of unsupervised domain adaptation is important and pertinent to the community.

2. The exposition on related techniques is quite comprehensive.

3. The ablation studies are thorough, and the experimental configurations are clearly presented.

**Weaknesses:**

1. While the authors' motivation offers some insight, the experimental results presented are not entirely convincing. Despite the inclusion of three carefully designed components, the model's performance only improves by 1% mIoU on the GTA to Cityscapes transfer. Moreover, the performance on the Synthia to Cityscapes transfer is even lower than the baseline (HRDA).

2. The authors only conducted experiments in the relatively simple and ideal scenario of transferring from synthetic datasets to real datasets, neglecting the more convincing Cityscapes to ACDC experiments. I believe that adding this experiment would make the authors' work more robust.

3. I have some reservations to the extent that the introduced three-branch and AAC modules may add additional parameters and computational load, potentially boosting the model's Oracle performance. The performance gains claimed by the authors could very well stem from this.

**Questions:**

See weaknesses above.

---

> ### Author Response · Authors · 2023-11-22
> **Response [1/2]**
>
> We thank the reviewer for the constructive feedback and provide responses below.
>
> **Q1: The performances gained on GTA5/Synthia $\to$ Cityscapes.**
>
> **A1:** We acknowledge the observed limitations in the performance gain of our method on the GTA5/Synthia $\to$ Cityscapes benchmarks. We believe two factors contribute to this result. Firstly, we utilized the same parameter settings as HRDA without fine-tuning for our method. Additionally, we employed mixed-precision during training to conserve graphics memory, which might have affected performance. We anticipate that further adjustments and application to GPU machines with larger memory capacities could lead to improved performance. Notably, after fine-tuning B3CT parameters, our results on the Synthia $\rightarrow$ Cityscapes benchmark have been updated from 65.8 mIoU to 67.0, as reflected in the revised Tab. 5.
>
> #### Comparison with existing methods on Synthia $\rightarrow$ Cityscapes
> || Road | S.walk | Build. | Wall | Fence | Pole | Tr.Light | Sign | Veget. | Terrain | Sky | Person | Rider | Car | Truck | Bus | Train | M.bike | Bike | mIoU |
> |-|-|-|-|-|-|-|-|-|-|-|-|-|-|-|-|-|-|-|-|-|
> | CBST | 68.0 | 29.9 | 76.3 | 10.8 | 1.4 | 33.9 | 22.8 | 29.5 | 77.6 | -- | 78.3 | 60.6 | 28.3 | 81.6 | -- | 23.5 | -- | 18.8 | 39.8 | 42.6 |
> | DACS | 80.6 | 25.1 | 81.9 | 21.5 | 2.9 | 37.2 | 22.7 | 24.0 | 83.7 | -- | 90.8 | 67.6 | 38.3 | 82.9 | -- | 38.9 | -- | 28.5 | 47.6 | 48.3 |
> | DPL-Dual | 83.5 | 38.2 | 80.4 | 1.3 | 1.1 | 29.1 | 20.2 | 32.7 | 81.8 | -- | 83.6 | 55.9 | 20.3 | 79.4 | -- | 26.6 | -- | 7.4 | 46.2 | 43.0 |
> | SAC | 89.3 | 47.2 | 85.5 | 26.5 | 1.3 | 43.0 | 45.5 | 32.0 | 87.1 | -- | 89.3 | 63.6 | 25.4 | 86.9 | -- | 35.6 | -- | 30.4 | 53.0 | 52.6 |
> | CorDA | **93.3**    | **61.6**    | 85.3 | 19.6 | 5.1 | 37.8 | 36.6 | 42.8 | 84.9 | -- | 90.4 | 69.7 | 41.8 | 85.6 | -- | 38.4 | -- | 32.6 | 53.9 | 55.0 |
> | ProDA    | 87.8 | 45.7 | 84.6 | 37.1 | 0.6 | 44.0 | 54.6 | 37.0 | 88.1 | -- | 84.4 | 74.2 | 24.3 | 88.2 | -- | 51.1 | -- | 40.5 | 45.6 | 55.5 |
> | DAFormer    | 84.5 | 40.7 | 88.4 | 41.5 | 6.5  | 50.0 | 55.0 | 54.6 | 86.0 | -- | 89.8 | 73.2 | 48.2 | 87.2 | -- | 53.2 | -- | 53.9 | 61.7 | 60.9 |
> | HRDA | 85.2 | 47.7 | **88.8**    | **49.5**    | 4.8 | 57.2 | 65.7 | 60.9 | 85.3 | -- | 92.9 | 79.4 | 52.8 | **89.0**    | -- | **64.7**    | -- | 63.9 | 64.9 | 65.8 |
> | DAFormer + B3CT | 82.8 | 44.2 | 86.7 | 38.9 | 5.0 | 55.0 | 63.0 | 61.2 | 83.6 | -- | 82.9 | 76.3 | 47.9 | 88.0 | -- | 58.6 | -- | 61.8 | 65.3 | 62.6 |
> | HRDA + B3CT | 90.5 | 57.5 | 88.7 | 46.3 | **7.5**     | **57.7**    | **66.6**    | **63.3**    | **89.1**    | -- | **94.2**    | **80.5**    | **55.8**    | **89.0**    | -- | 54.4 | -- | **64.6**    | **65.7**    | **67.0**  |
>
> **Q2: Experiments on Cityscapes $\to$ ACDC benchmark.**
>
> **A2:** To assess the generalization capabilities of our proposed method across diverse scenarios, we conducted additional validation on the clear-to-adverse-weather Cityscapes $\to$ ACDC benchmark, which involves domain adaptation from clear weather conditions in Cityscapes to adverse weather conditions. Our re-implementation of HRDA achieves 64.3 mIoU on the test set, which is basically consistent with third-party re-implementation results[1]. In comparison to this, B3CT further enhances performance to 66.9 mIoU. These experimental results prove the robustness of our B3CT. More detailed experimental results can be found in the table below and App. A.1.
>
> [1] CDAC: Cross-domain Attention Consistency in Transformer for Domain Adaptive Semantic Segmentation, ICCV 2023.
>
> #### Comparison with existing methods on Cityscapes $\rightarrow$ ACDC
>
> || Road | S.walk | Build. | Wall | Fence | Pole | Tr.Light | Sign | Veget. | Terrain | Sky | Person | Rider | Car | Truck | Bus | Train | M.bike | Bike | mIoU |
> |----|-|-|-|-|-|-|-|-|-|-|-|-|-|-|-|-|-|-|-|-|
> | ADVENT | 72.9 | 14.3 | 40.5 | 16.6 | 21.2 | 9.3 | 17.4 | 21.2 | 63.8 | 23.8 | 18.3 | 32.6 | 19.5 | 69.5 | 36.2 | 34.5 | 46.2 | 26.9 | 36.1 | 32.7 |
> | MGCDA | 73.4 | 28.7 | 69.9 | 19.3 | 26.3 | 36.8 | 53.0 | 53.3 | **75.4**    | 32.0 | 84.6 | 51.0 | 26.1 | 77.6 | 43.2 | 45.9 | 53.9 | 32.7 | 41.5 | 48.7 |
> | DANNet    | 84.3 | 54.2 | 77.6 | 38.0 | 30.0 | 18.9 | 41.6 | 35.2 | 71.3 | 39.4 | **86.6**    | 48.7 | 29.2 | 76.2 | 41.6 | 43.0 | 58.6 | 32.6 | 43.9 | 50.0 |
> | DAFormer | 58.4 | 51.3 | 84.0 | 42.7 | 35.1 | 50.7 | 30.0 | 57.0 | 74.8 | 52.8 | 51.3 | 58.3 | 32.6 | 82.7 | 58.3 | 54.9 | 82.4 | 44.1 | 50.7 | 55.4 |
> | HRDA | 69.1 | **64.8** | 86.4 | **53.9**    | 40.3 | **56.5**    | 41.1 | **63.9**    | 72.8 | **58.8**    | 62.0 | **69.9**    | 44.2 | 87.0 | **73.5**    | 82.0 | 87.8 | 45.4 | **63.0**    | 64.3 |
> | HRDA + B3CT | **89.8**    | 62.0 | **86.6**    | 53.5 | **40.7**    | 48.9 | **58.8**    | 60.1 | 73.2 | 57.1 | 85.7 | 67.9 | **44.6**    | **87.9**    | 72.5 | **84.3**    | **88.4**    | **51.6**    | 56.8 | **66.9**    |

---

> > ### Author Response · Authors · 2023-11-22
> > **Response [2/2]**
> >
> > **Q3: The relationship between parameter quantity and model performance.**
> >
> > **A3:** We attribute the improvement in model performance to the training strategy that selectively aligns features between different domains. **Firstly**, the hybrid branch is only added during the training phase. B3CT and HRDA have identical testing processes where no additional parameters are added.
> > **Secondly**, our model only brought an increase of 0.4M parameters during the training phase, which can be ignored compared to the overall parameters (86.1M). We have included more detailed supplements in the table below and appendix.
> >
> > #### Comparison of computational efficiency
> >
> > | UDA method  | Throughput(img/s) | TFLOPS | Training Params. |
> > |-|-|-|-|
> > | DAFormer    | 0.7 | 1.6    | 85.2M   |
> > | HRDA | 0.8 | 3.3    | 85.7M   |
> > | HRDA + B3CT | 0.8 | 3.3    | 86.1M   |
> >
> > Thank you for your thoughtful comments and suggestions again.

---

### Official Review · Reviewer_YWN9 · 2023-10-31

**Soundness:** 3 good
**Presentation:** 3 good
**Contribution:** 2 fair
**Rating:** 5
**Confidence:** 3

**Summary:**

This paper introduces an unsupervised domain adaptive segmentation method named B$^3$CT. In addition to the traditional UDA learning approach, which involves supervised training on the source domain and re-training on the target domain, the key idea lies in the introduction of a third branch that facilitates the alignment of the source and target domains through cross-attention mechanisms.

When addressing the question of "where to align," the paper introduces an Adaptive Alignment Controller (AAC) at each layer to determine varying degrees of alignment. As for the question of "when to align," the paper defines a coordinate weight that controls the loss value within the hybrid branch, where the coordinate weight is derived from the pseudo-accuracy of target predictions generated by the student model compared to the target pseudo-labels provided by the teacher model. The experiments are conducted on two public benchmarks: GTA5-to-CityScapes and Synthia-to-CityScapes, showing the effectiveness of the method.

**Strengths:**

- The paper is well-written and easy to follow.
- The concept of using cross-attention to align source and target features is elegant and logically sound.
- The design of the coordinate weight, which operates smoothly, is well-founded.

**Weaknesses:**

- The paper claims to achieve state-of-the-art performance on GTAV→Cityscapes (74.8), which is not entirely accurate. A previously published paper, MIC (CVPR2023) [1], has achieved a higher performance of 75.9. I noticed the authors did not cite and compare it. Therefore, the claimed contribution should be reconsidered.
- The proposed B$^3$CT is only applicable to transformer-based architectures and cannot be employed with CNN-based models.
- B$^3$CT introduces additional computation during the inference stage, which raises concerns about computational efficiency for both training and testing. It would be valuable to include a comparison of computational efficiency in the paper's evaluation and report the results accordingly.

[1] MIC: Masked Image Consistency for Context-Enhanced Domain Adaptation. CVPR2023.

**Questions:**

Will authors release code?

---

> ### Author Response · Authors · 2023-11-22
> **Response**
>
> We thank the reviewer for the constructive feedback and provide responses below.
>
> **Q1: Inappropraite claim of contribution.**
>
> **A1:** We apologize for the inaccurate description of the state-of-the-art information. We have modified the statement regarding performance. At present, experiments based on HRDA and DAFormer baseline have verified the effectiveness of B3CT. The results are shown as follows. In the future, we will apply this method to more baselines including MIC.
>
> #### Comparison with existing methods on GTA5 $\rightarrow$ Cityscapes
> || Road | S.walk | Build. | Wall | Fence | Pole | Tr.Light | Sign | Veget. | Terrain | Sky | Person | Rider | Car | Truck | Bus | Train | M.bike | Bike | mIoU |
> |-|-|-|-|-|-|-|-|-|-|-|-|-|-|-|-|-|-|-|-|-|
> | DAFormer    | 95.7 | 70.2 | 89.4 | 53.5 | 48.1 | 49.6 | 55.8 | 59.4 | 89.9 | 47.9 | 92.5 | 72.2 | 44.7 | 92.3 | 74.5 | 78.2 | 65.1 | 55.9 | 61.8 | 68.3 |
> | HRDA | 96.4 | 74.4 | 91.0 | **61.6**    | 51.5 | 57.1 | 63.9 | **69.3**    | 91.3 | 48.4 | 94.2 | 79.0 | 52.9 | 93.9 | **84.1**    | 85.7 | 75.9 | **63.9**    | 67.5 | 73.8 |
> | DAFormer + B3CT | **96.5**    | **75.2**    | 90.7 | 60.6 | 45.5 | 57.5 | 63.7 | 68.6 | 90.9 | 48.4 | 91.9 | 76.7 | 49.2 | 92.1 | 59.5 | 68.3 | 64.6 | 58.7 | 67.9 | 69.8 |
> | HRDA + B3CT | **96.5**    | 75.0 | **91.2**    | 59.6 | **55.4**    | **58.5**    | **66.3**    | 69.1 | **91.9**    | **49.3**    | **94.4**    | **79.2**    | **55.3**    | **94.2**    | 83.2 | **88.7**    | **80.3**    | 63.7 | **68.4**    | **74.8**   |
>
> #### Comparison with existing methods on Synthia $\rightarrow$ Cityscapes
> || Road | S.walk | Build. | Wall | Fence | Pole | Tr.Light | Sign | Veget. | Terrain | Sky | Person | Rider | Car | Truck | Bus | Train | M.bike | Bike | mIoU |
> |-|-|-|-|-|-|-|-|-|-|-|-|-|-|-|-|-|-|-|-|-|
> | DAFormer    | 84.5 | 40.7 | 88.4 | 41.5 | 6.5  | 50.0 | 55.0 | 54.6 | 86.0 | -- | 89.8 | 73.2 | 48.2 | 87.2 | -- | 53.2 | -- | 53.9 | 61.7 | 60.9 |
> | HRDA | 85.2 | 47.7 | **88.8**    | **49.5**    | 4.8 | 57.2 | 65.7 | 60.9 | 85.3 | -- | 92.9 | 79.4 | 52.8 | **89.0**    | -- | **64.7**    | -- | 63.9 | 64.9 | 65.8 |
> | DAFormer + B3CT | 82.8 | 44.2 | 86.7 | 38.9 | 5.0 | 55.0 | 63.0 | 61.2 | 83.6 | -- | 82.9 | 76.3 | 47.9 | 88.0 | -- | 58.6 | -- | 61.8 | 65.3 | 62.6 |
> | HRDA + B3CT | 90.5 | 57.5 | 88.7 | 46.3 | **7.5**     | **57.7**    | **66.6**    | **63.3**    | **89.1**    | -- | **94.2**    | **80.5**    | **55.8**    | **89.0**    | -- | 54.4 | -- | **64.6**    | **65.7**    | **67.0**  |
>
> **Q2: Applicability of the proposed method to CNN-based models**
>
> **A2:** Currently, the best methods for UDA segmentation are based on transformers. Therefore, our starting point is to optimize the attention mechanism in transformers. To be more specific, we introduce cross-attention based on the transformer's self-attention to achieve domain alignment. Except for the AAC module, which introduces very few parameters, different attention calculations use exactly the same network weights. Therefore, it can be easily applied to various transformer backbones, such as HRDA and DAFormer. However, because CNN's backbone does not contain the attention mechanism, it is necessary to design other modules and introduce additional adaptations to apply our idea. Thank you for your insightful input, and we will be committed to improving the adaptability of our approach to a wider range of backbone architectures.
>
> **Q3: Comparison of computational efficiency.**
>
> **A3:** In response to the inquiry about computational efficiency, we have provided additional information in the appendix regarding the runtime and parameters of HRDA during inference and demonstrate it below. **Firstly**, the hybrid branch is only added during the training phase. It has no adverse impact on inference speed during testing. Therefore, our model has the same throughput (0.8) and TFLOPs (3.3) as HRDA.
> **Secondly**, our model only brought an increase of 0.4M parameter quantity during the training phase, which can be ignored compared to the overall parameter quantity (86.1M).  In summary, our method achieves competitive performance while upholding computational efficiency and ensuring swift inference speed.
>
> #### Comparison of computational efficiency
>
> | UDA method  | Throughput(img/s) | TFLOPS | Training Params. |
> |-|-|-|-|
> | DAFormer    | 0.7 | 1.6    | 85.2M   |
> | HRDA | 0.8 | 3.3    | 85.7M   |
> | HRDA + B3CT | 0.8 | 3.3    | 86.1M   |
>
> **Q4: Access to the Code.**
>
> **A4:** The code will be released promptly after acceptance.
>
> Thank you for your thoughtful comments and suggestions again.

---

### Official Review · Reviewer_qoMd · 2023-11-01

**Soundness:** 3 good
**Presentation:** 2 fair
**Contribution:** 3 good
**Rating:** 5
**Confidence:** 5

**Summary:**

This paper focuses on unsupervised domain adaptation and proposes a multi-branch coordinated training method. Specifically, it designs three-branch coordinated training technique, where the final loss function is dynamically weighted by coordinate weight on the loss values of three branches. Extensive experiments show that the proposed method has achieved good unsupervised domain adaptation performance.

**Strengths:**

-a. It proposes a multi-branch coordinated training method for unsupervised domain adaptation.
- b. It designs three-branch coordinated training technique, where the final loss function is dynamically weighted by coordinate weight on the loss values of three branches.
- c. The experimental results show that the performance of the proposed method is promising in unsupervised domain adaptation.

**Weaknesses:**

Although this paper is well written with comprehensive evaluation and good results, there are still some issues.
Several parts of this paper are not very clear and need further clarification. Please check the questions.
In addition, some key related works that also address unsupervised domain adaptation are missed. Please check the questions.
1. In table 5, it seems that the proposed method performs less effectively on Synthia to Cityscapes benchmark. It would be better to provide some insights and analysis to illustrate these results.
2. As shown in the ablation studies in Tables 1-3, the performance gains seem not significant. Did the author conduct experiment with multiple random runs or random seeds? It is not clear how much the training randomness affects the performance when the gains are not significant.
3. The key related unsupervised domain adaptation papers [A, B, C, D] are missed. This paper focuses on unsupervised domain adaptation and proposes a multi-branch coordinated training method. The differences, pros and cons of the proposed multi-branch coordinated training method and the traditional UDA co-training methods [A, B, C, D] are not clear. [A] introduces multiple feature spaces and performs co-training by conducting Co-regularized Alignment among them, whereas [B] introduces multiple classifiers and performs co-training by conducting Collaborative Alignment upon them. [C] achieves co-training by introducing multiple diverse classifiers to generate class-balance weights, which are then used to weight/regularize adversarial learning or self-training, whereas [B] achieves co-training by leveraging current and historical models to generate historical consistency weights, which are then used to weight/regularize adversarial learning or self-training. It would be better to provide discussion and analysis to illustrate the differences, pros and cons of the proposed method and [A,B,C,D]. For example, the differences, pros and cons of the proposed coordinate weights, the class-balance weights in [C] and the historical consistency weights in [D], for UDA.
[A] Co-regularized Alignment for Unsupervised Domain Adaptation. NIPS 2018
[B] Collaborative and Adversarial Network for Unsupervised domain adaptation. CVPR 2018
[C] Taking A Closer Look at Domain Shift: Category-level Adversaries for Semantics Consistent Domain Adaptation. CVPR 2019.
[D] Model Adaptation: Historical Contrastive Learning for Unsupervised Domain Adaptation without Source Data. NeurIPS 2021.
4. According to Table 3, it seems that HRDA (73.8) has been taken as the baseline. Did the author try to apply the proposed method on other methods/baselines and what about the performances/gains? It is very interesting to investigate the generalization ability of the proposed method by testing it on other baselines.

**Questions:**

Please check Weaknesses.

Conclusion
Overall, this work proposes a multi-branch coordinated training method for UDA and yields good experimental results. However, there are some details that need to be made clearer, as listed in the questions. I would like to upgrade the score if the questions could be well addressed.

---

> ### Author Response · Authors · 2023-11-22
> **Response [1/2]**
>
> We thank the reviewer for the constructive feedback and provide responses below.
>
> **Q1: Experiments on Synthia $\rightarrow$ Cityscapes benchmark.**
>
> **A1:** On the two benchmarks, the previous training of B3CT used the same experimental settings as HRDA without making any changes. We have made appropriate adjustments to the parameters for the design of B3CT and increased the original 65.8 mIoU to 67.0 on the Synthia $\rightarrow$ Cityscapes benchmark. Tab. 5 has been updated to the latest performance.
>
> **Q2: The influence of randomness in experiments.**
>
> **A2:** Thank you for raising this valuable question. Randomness does indeed have a certain impact on the experimental results. We did not include the impact of randomness in the table to better compare it with previous work. Actually, all of the results are averaged over 3 random seeds. We have added this experimental detail in Sec.4. To demonstrate the impact of randomness, we present the variance of the experimental data obtained in Tab. 5 below.
>
> #### Comparison with existing methods on GTA5 $\rightarrow$ Cityscapes
> || Road | S.walk | Build. | Wall | Fence | Pole | Tr.Light | Sign | Veget. | Terrain | Sky | Person | Rider | Car | Truck | Bus | Train | M.bike | Bike | mIoU |
> |-|-|-|-|-|-|-|-|-|-|-|-|-|-|-|-|-|-|-|-|-|
> | CBST | 91.8 | 53.5 | 80.5 | 32.7 | 21.0 | 34.0 | 28.9 | 20.4 | 83.9 | 34.2 | 80.9 | 53.1 | 24.0 | 82.7 | 30.3 | 35.9 | 16.0 | 25.9 | 42.8 | 45.9 |
> | DACS | 89.9 | 39.7 | 87.9 | 30.7 | 39.5 | 38.5 | 46.4 | 52.8 | 88.0 | 44.0 | 88.8 | 67.2 | 35.8 | 84.5 | 45.7 | 50.2 | 0.0 | 27.3 | 34.0 | 52.1 |
> | DPL-Dual    | 92.8 | 54.4 | 86.2 | 41.6 | 32.7 | 36.4 | 49.0 | 34.0 | 85.8 | 41.3 | 86.0 | 63.2 | 34.2 | 87.2 | 39.3 | 44.5 | 18.7 | 42.6 | 43.1 | 53.3 |
> | SAC | 90.4 | 53.9 | 86.6 | 42.4 | 27.3 | 45.1 | 48.5 | 42.7 | 87.4 | 40.1 | 86.1 | 67.5 | 29.7 | 88.5 | 49.1 | 54.6 | 9.8 | 26.6 | 45.3 | 53.8 |
> | CorDA | 94.7 | 63.1 | 87.6 | 30.7 | 40.6 | 40.2 | 47.8 | 51.6 | 87.6 | 47.0 | 89.7 | 66.7 | 35.9 | 90.2 | 48.9 | 57.5 | 0.0 | 39.8 | 56.0 | 56.6 |
> | ProDA | 87.8 | 56.0 | 79.7 | 46.3 | 44.8 | 45.6 | 53.5 | 53.5 | 88.6 | 45.2 | 82.1 | 70.7 | 39.2 | 88.8 | 45.5 | 59.4 | 1.0 | 48.9 | 56.4 | 57.5 |
> | DAFormer    | 95.7 | 70.2 | 89.4 | 53.5 | 48.1 | 49.6 | 55.8 | 59.4 | 89.9 | 47.9 | 92.5 | 72.2 | 44.7 | 92.3 | 74.5 | 78.2 | 65.1 | 55.9 | 61.8 | 68.3 |
> | HRDA | 96.4 | 74.4 | 91.0 | **61.6**    | 51.5 | 57.1 | 63.9 | **69.3**    | 91.3 | 48.4 | 94.2 | 79.0 | 52.9 | 93.9 | **84.1**    | 85.7 | 75.9 | **63.9**    | 67.5 | 73.8 |
> | DAFormer + B3CT | **96.5**    | **75.2**    | 90.7 | 60.6 | 45.5 | 57.5 | 63.7 | 68.6 | 90.9 | 48.4 | 91.9 | 76.7 | 49.2 | 92.1 | 59.5 | 68.3 | 64.6 | 58.7 | 67.9 | 69.8 ± 0.4 |
> | HRDA + B3CT | **96.5**    | 75.0 | **91.2**    | 59.6 | **55.4**    | **58.5**    | **66.3**    | 69.1 | **91.9**    | **49.3**    | **94.4**    | **79.2**    | **55.3**    | **94.2**    | 83.2 | **88.7**    | **80.3**    | 63.7 | **68.4**    | **74.8** ± 0.3   |
>
> #### Comparison with existing methods on Synthia $\rightarrow$ Cityscapes
> || Road | S.walk | Build. | Wall | Fence | Pole | Tr.Light | Sign | Veget. | Terrain | Sky | Person | Rider | Car | Truck | Bus | Train | M.bike | Bike | mIoU |
> |-|-|-|-|-|-|-|-|-|-|-|-|-|-|-|-|-|-|-|-|-|
> | CBST | 68.0 | 29.9 | 76.3 | 10.8 | 1.4 | 33.9 | 22.8 | 29.5 | 77.6 | -- | 78.3 | 60.6 | 28.3 | 81.6 | -- | 23.5 | -- | 18.8 | 39.8 | 42.6 |
> | DACS | 80.6 | 25.1 | 81.9 | 21.5 | 2.9 | 37.2 | 22.7 | 24.0 | 83.7 | -- | 90.8 | 67.6 | 38.3 | 82.9 | -- | 38.9 | -- | 28.5 | 47.6 | 48.3 |
> | DPL-Dual | 83.5 | 38.2 | 80.4 | 1.3 | 1.1 | 29.1 | 20.2 | 32.7 | 81.8 | -- | 83.6 | 55.9 | 20.3 | 79.4 | -- | 26.6 | -- | 7.4 | 46.2 | 43.0 |
> | SAC | 89.3 | 47.2 | 85.5 | 26.5 | 1.3 | 43.0 | 45.5 | 32.0 | 87.1 | -- | 89.3 | 63.6 | 25.4 | 86.9 | -- | 35.6 | -- | 30.4 | 53.0 | 52.6 |
> | CorDA | **93.3**    | **61.6**    | 85.3 | 19.6 | 5.1 | 37.8 | 36.6 | 42.8 | 84.9 | -- | 90.4 | 69.7 | 41.8 | 85.6 | -- | 38.4 | -- | 32.6 | 53.9 | 55.0 |
> | ProDA    | 87.8 | 45.7 | 84.6 | 37.1 | 0.6 | 44.0 | 54.6 | 37.0 | 88.1 | -- | 84.4 | 74.2 | 24.3 | 88.2 | -- | 51.1 | -- | 40.5 | 45.6 | 55.5 |
> | DAFormer    | 84.5 | 40.7 | 88.4 | 41.5 | 6.5  | 50.0 | 55.0 | 54.6 | 86.0 | -- | 89.8 | 73.2 | 48.2 | 87.2 | -- | 53.2 | -- | 53.9 | 61.7 | 60.9 |
> | HRDA | 85.2 | 47.7 | **88.8**    | **49.5**    | 4.8 | 57.2 | 65.7 | 60.9 | 85.3 | -- | 92.9 | 79.4 | 52.8 | **89.0**    | -- | **64.7**    | -- | 63.9 | 64.9 | 65.8 |
> | DAFormer + B3CT | 82.8 | 44.2 | 86.7 | 38.9 | 5.0 | 55.0 | 63.0 | 61.2 | 83.6 | -- | 82.9 | 76.3 | 47.9 | 88.0 | -- | 58.6 | -- | 61.8 | 65.3 | 62.6 ± 1.0 |
> | HRDA + B3CT | 90.5 | 57.5 | 88.7 | 46.3 | **7.5**     | **57.7**    | **66.6**    | **63.3**    | **89.1**    | -- | **94.2**    | **80.5**    | **55.8**    | **89.0**    | -- | 54.4 | -- | **64.6**    | **65.7**    | **67.0** ± 0.4  |

---

> > ### Author Response · Authors · 2023-11-22
> > **Response [2/2]**
> >
> > **Q3: Missing key-related UDA papers.**
> >
> > **A3:** Thank you for listing the relevant papers. We believe these works are a good supplement to our approach.
> > - In Sec. 2, the existing UDA domain alignment methods are divided into pixel, prototype, and label level alignments. The paper **[A]Co-regularized Alignment for Unsupervised Domain Adaptation** is a good supplement to the label-level alignment work, which aligns the prediction results of different networks with multiple feature spaces. We have added it to the related work.
> > - The paper **[B]Collaborative and Adversarial Network for Unsupervised Domain Adaptation** can better explain the original intention and experimental results of our AAC module design. Aiming at learning domain informative representation at lower layers and domain uninformative representation at higher layers, CAN proposed a collaborative and adversarial learning scheme. However, the optimization weights learned by CAN can only be fixed for different network layers, and cannot consider fine-grained semantic information in semantic segmentation tasks. We discussed this work in Sec. 3.2.
> > - **[C]Taking A Closer Look at Domain Shift: Category level Adversaries for Semantics Consistent Domain Adaptation** and **[D]Model Adaptation: Historical Comparative Learning for Unsupervised Domain Adaptation without Source Data** have both dynamically adjusted the weight of the loss function based on the model prediction results. The main idea is that when the prediction results of different classifiers or time steps differ significantly, it can be considered that the current sample or network layer needs to focus more on weight updates. However, we argue that a well-performing student model means that the teacher model can provide reliable pseudo-labels. Only when the output of the student model and the teacher model are relatively consistent, can we introduce hybrid computation into the network. We have added a comparative discussion of these two tasks in section 3.3. Thank you again for your suggestion.
> >
> > **Q4: Further comparison on different baseline/method.**
> >
> > **A4:** Investigating the generalization ability of a model is of great importance. We applied the B3CT model using DAFormer as the backbone on two benchmarks, GTA5 $\rightarrow$ Cityscapes and Synthia $\rightarrow$ Cityscapes respectively. The experimental results are shown in the table below and updated in Tab. 5. B3CT achieves a 1.5 and 1.7 percent point improvement over the respective two benchmarks. This implies that B3CT has the flexibility to integrate with various Transformer-based models seamlessly.
> >
> > #### Further baseline comparison on GTA5 $\rightarrow$ Cityscapes
> > || Road | S.walk | Build. | Wall | Fence | Pole | Tr.Light | Sign | Veget. | Terrain | Sky | Person | Rider | Car | Truck | Bus | Train | M.bike | Bike | mIoU |
> > |-|-|-|-|-|-|-|-|-|-|-|-|-|-|-|-|-|-|-|-|-|
> > | DAFormer    | 95.7 | 70.2 | 89.4 | 53.5 | 48.1 | 49.6 | 55.8 | 59.4 | 89.9 | 47.9 | 92.5 | 72.2 | 44.7 | 92.3 | 74.5 | 78.2 | 65.1 | 55.9 | 61.8 | 68.3 |
> > | HRDA | 96.4 | 74.4 | 91.0 | **61.6**    | 51.5 | 57.1 | 63.9 | **69.3**    | 91.3 | 48.4 | 94.2 | 79.0 | 52.9 | 93.9 | **84.1**    | 85.7 | 75.9 | **63.9**    | 67.5 | 73.8 |
> > | DAFormer + B3CT | **96.5**    | **75.2**    | 90.7 | 60.6 | 45.5 | 57.5 | 63.7 | 68.6 | 90.9 | 48.4 | 91.9 | 76.7 | 49.2 | 92.1 | 59.5 | 68.3 | 64.6 | 58.7 | 67.9 | 69.8 ± 0.4 |
> > | HRDA + B3CT | **96.5**    | 75.0 | **91.2**    | 59.6 | **55.4**    | **58.5**    | **66.3**    | 69.1 | **91.9**    | **49.3**    | **94.4**    | **79.2**    | **55.3**    | **94.2**    | 83.2 | **88.7**    | **80.3**    | 63.7 | **68.4**    | **74.8** ± 0.3   |
> >
> > #### Further baseline comparison on Synthia $\rightarrow$ Cityscapes
> > || Road | S.walk | Build. | Wall | Fence | Pole | Tr.Light | Sign | Veget. | Terrain | Sky | Person | Rider | Car | Truck | Bus | Train | M.bike | Bike | mIoU |
> > |-|-|-|-|-|-|-|-|-|-|-|-|-|-|-|-|-|-|-|-|-|
> > | DAFormer    | 84.5 | 40.7 | 88.4 | 41.5 | 6.5  | 50.0 | 55.0 | 54.6 | 86.0 | -- | 89.8 | 73.2 | 48.2 | 87.2 | -- | 53.2 | -- | 53.9 | 61.7 | 60.9 |
> > | HRDA | 85.2 | 47.7 | **88.8**    | **49.5**    | 4.8 | 57.2 | 65.7 | 60.9 | 85.3 | -- | 92.9 | 79.4 | 52.8 | **89.0**    | -- | **64.7**    | -- | 63.9 | 64.9 | 65.8 |
> > | DAFormer + B3CT | 82.8 | 44.2 | 86.7 | 38.9 | 5.0 | 55.0 | 63.0 | 61.2 | 83.6 | -- | 82.9 | 76.3 | 47.9 | 88.0 | -- | 58.6 | -- | 61.8 | 65.3 | 62.6 ± 1.0 |
> > | HRDA + B3CT | 90.5 | 57.5 | 88.7 | 46.3 | **7.5**     | **57.7**    | **66.6**    | **63.3**    | **89.1**    | -- | **94.2**    | **80.5**    | **55.8**    | **89.0**    | -- | 54.4 | -- | **64.6**    | **65.7**    | **67.0** ± 0.4  |
> >
> > Thank you for your thoughtful comments and suggestions again.

---

### Meta-Review · Area_Chair_mDfE · 2023-12-05

**Metareview:**

This paper presents an unsupervised domain adaptation (UDA) method for semantic segmentation. It used an additional branch to align the source and target features.

After the rebuttal and AC-reviewer discussion stage, the final scores of this paper are 3/5/5/5. In the discussion stage, only one reviewer (rating 3) responded and insisted on rejection. The AC read the reviews and the responses, and found no reason to overturn the reviewers' comments.

**Justification For Why Not Higher Score:**

The average score falls below the acceptance threshold.

**Justification For Why Not Lower Score:**

N/A

---

### Decision · Program_Chairs · 2024-01-16

Reject